# Prediction of Fetal Blood Pressure during Labour with Deep Learning Techniques

**DOI:** 10.3390/bioengineering10070775

**Published:** 2023-06-28

**Authors:** John Tolladay, Christopher A. Lear, Laura Bennet, Alistair J. Gunn, Antoniya Georgieva

**Affiliations:** 1Oxford Labour Monitoring Group, Nuffield Department of Women’s and Reproductive Health, University of Oxford, Oxford, OX1 2JD, UK; john.tolladay@wrh.ox.ac.uk; 2The Fetal Physiology and Neuroscience Group, Department of Physiology, University of Auckland, Auckland 1010, New Zealand; christopher.lear@auckland.ac.nz (C.A.L.); l.bennet@auckland.ac.nz (L.B.); aj.gunn@auckland.ac.nz (A.J.G.); 3Big Data Institute, Old Road Campus, University of Oxford, Oxford, OX3 7LF, UK

**Keywords:** umbilical occlusions, cardiotocography, blood pressure, electronic fetal monitoring

## Abstract

Our objective is to develop a model for the prediction of minimum fetal blood pressure (FBP) during fetal heart rate (FHR) decelerations. Experimental data from umbilical occlusions in near-term fetal sheep (2698 occlusions from 57 near-term lambs) were used to train a convolutional neural network. This model was then used to estimate FBP for decelerations extracted from the final 90 min of 53,445 human FHR signals collected using cardiotocography. Minimum sheep FBP was predicted with a mean absolute error of 6.7 mmHg (25th, 50th, 75th percentiles of 2.3, 5.2, 9.7 mmHg), mean absolute percentage errors of 17.3% (5.5%, 12.5%, 23.9%) and a coefficient of determination $R^{2}=0.36$. While the model was unable to clearly predict severe compromise at birth in humans, there is positive evidence that such a model could predict human FBP with further development. The neural network is capable of predicting FBP for many of the sheep decelerations accurately but performed far from satisfactory at identifying FHR segments that correspond to the highest or lowest minimum FBP. These results indicate that with further work and a larger, more variable training dataset, the model could achieve higher accuracy.

## 1. Introduction

There is currently no practical and non-invasive method of continuously measuring fetal blood pressure (FBP) during labour. Non-invasive methods of detecting FBP using Doppler ultrasound- and computational fluid dynamic-based techniques have been developed (see [1,2,3,4] for example), but their practicality is limited during labour due to the position of the fetus and fetal/maternal movements. Any such device will face considerable challenges in reading the blood flow through an artery within a moving fetus undergoing labour. The key research gap is that fetal blood pressure and the onset of fetal hypotension cannot be routinely measured in human labour.

The fetal heart rate (FHR) is a far less challenging biomarker to measure and is routinely recorded using cardiotocography (CTG). Pre-clinical studies in fetal sheep have utilised repetitive labor-like umbilical cord occlusions, causing severe fetal hypotension and compromise, but unfortunately showed that FHR patterns were poor indicators of fetal hypotension [5]. Nonetheless, Bennet et al. [6] showed that fetal compromise was associated with subtle changes, including an increase in the slope and magnitude of the initial deceleration of the FHR with fetal hypotension. It has been proposed that the depth, duration and frequency of intrapartum decelerations of the FHR represent the best indicators of hypoxaemia and acidaemia, but it is clear that additional biomarkers that more closely reflecting the onset of hypotension must be sought [7]. Higher deceleration area and decelerative capacity, as calculated by automated, computer-based methods, have also been associated with hypotension in fetal sheep [8]. Additional work has also suggested that in-depth analysis of FHR patterns may provide early warning of fetal cardiovascular decompensation and acidemia [9,10,11].

Continuous fetal monitoring during labour using CTG is known to have many limitations and only provides a proxy measurement of fetal oxygenation. Blood flow is determined partly by blood pressure and, in turn, by cardiac contractility and vascular resistance. It is now well established that brain perfusion is critically compromised when fetal blood pressure falls below baseline values. Thus, hypotension during severe or recurrent hypoxia critically compromises cerebral perfusion and precipitates hypoxic-ischaemic injury across multiple animal models. Fetal blood pressure monitoring would provide a more direct measure of blood flow reaching the brain, particularly during contractions, when FBP may dip intermittently to dangerously low levels. Our aim was to investigate the relationship between FHR and FBP during decelerations, seeking to produce deep learning models to predict/infer the minimal FBP value based solely on the FHR trace recorded with standard CTG fetal monitors. If successful, this would enable medical practitioners to monitor the blood pressure of fetuses non-invasively during labour.

Sheep have been used for over 80 years to study aspects of pregnancy that are impractical or unethical to study in humans. Initially, they were used to study umbilical blood flow and transfer between fetal sheep and their mothers [12,13]. Sheep have been recognised as an imperfect model for human pregnancy, but have many commonalities with humans in this regard. The neural and cardiovascular maturation of near-term fetal sheep at 0.85 of gestation (≈125 days, the term is 147 days) is broadly equivalent to term human neonates [14,15,16]. Multiple parallels between the FHR patterns in near-term fetal sheep experiments and human labour have indeed been observed [8,17,18]. It is also possible to sample from both the maternal and fetal circulations under steady-state conditions repetitively and to take measurements, such as blood pressure, making pregnant sheep an invaluable model for the study of fetal physiology [19]. [20] provide further details regarding the similarities between sheep and human pregnancies.

Deep learning methods have been applied to CTG records previously, but such studies have tended to use FHR to predict the severity of birth outcomes (see for example [21,22,23,24]). In this study, we took a novel approach and applied state-of-the-art deep learning methods to the Auckland experimental data to model a relationship between FHR and the corresponding minima in FBP. We then applied the model to the Oxford data to examine whether there was a relationship between predicted FBP and the incidence of severe neonatal compromise at birth. Deep learning methods are able to automatically extract and utilise relevant features from the data on which they are trained to improve their predictions. These methods may, therefore, be able to find extra biomarkers that have not yet been found by researchers. The data and model configuration and testing are described in Section 2 with the results of the final model discussed in Section 3 and concluding remarks given in Section 4.

## 2. Methods

This work builds on two main datasets: (a) experimental data from Auckland providing continuous and concurrent FHR and FBP signals from 57 instrumented sheep undergoing repeated, complete umbilical cord occlusions to simulate contractions during labour and (b) routinely collected data from Oxford’s delivery ward from over 50,000 term births monitored with standard CTG during labour.

### 2.1. Experimental Data for the Fetal Sheep

The Auckland dataset [5,6,8,20,25,26,27] consists of 1 Hz fetal heart rate and blood pressure traces from 57 instrumented sheep divided into four experimental groups, as detailed in Table 1. This includes eight chronically hypoxic sheep, for which current methods find it more difficult to predict hypotension [28].

Fetal sheep experiments were performed as previously described, with full details given by Lear et al. [20]. Briefly, fetuses were surgically instrumented under general anaesthesia. Femoral artery catheters and subcutaneous electrodes over the right shoulder and left fifth-intercostal space were placed to measure FBP (sampled at 64 Hz) and fetal electrocardiogram (sampled at 1024 Hz), respectively. An inflatable silicone occluder was placed around the umbilical cord (18HD, in vivo Metric, Healdsburg, CA, USA). Fetal leads were exteriorised through the maternal flank and a maternal long saphenous vein was catheterised for post-operative care. Maternal incisions were infiltrated with a long-acting analgesic, bupivacaine plus adrenaline (AstraZeneca). Ewes were revived from anaesthesia and housed together in separate metabolic cages with ad libitum access to food and water, in temperature-controlled rooms (16±1 °C, humidity 50±10%) with a 12 h light/dark cycle. Daily intravenous antibiotics were administered to the ewe for 4 days (600 mg of benzylpenicillin sodium; Novartis, Auckland, New Zealand and 80 mg of gentamicin).

Experiments began 4 to 5 days after surgery at 0.85 of gestation. Normoxic fetuses (PaO_2_
≥17 mmHg) were randomly assigned to groups N1-5, N1-2.5 or N2-2, as defined in Table 1; fetuses with stable chronic hypoxaemia (PaO_2_
<17 mmHg for ≥3 days) were assigned to the H1-5 group. No animals were in labour during these experiments. Umbilical cord occlusions were induced by rapid inflation of the umbilical cord occluder with a volume of saline known to completely occlude the umbilical cord. At the end of the occlusion, the occluder was rapidly and completely deflated. After a period of reperfusion, the process was repeated continually for a maximum of 4 hours or until MAP fell below 20 mmHg on two successive occlusions. The duration of occlusions and reperfusion in each group are shown in Table 1. Ewes and fetuses were killed after the end of the experiments by an intravenous overdose of pentobarbital sodium administered to the ewe (9 g, Chemstock International, Christchurch, New Zealand).

### 2.2. Extracting the Fetal Sheep Data

The start time of the first occlusion for each sheep was located manually and subsequent occlusions were located at regular spacings of 2.5 or 5 min, depending on the experimental group. The FHR signal was then sliced into 150 s segments, each containing a deceleration of heart rate corresponding to the start of each umbilical occlusion and a subsequent portion of the recovery period. Where more than 50 s of a segment was missing due to data loss (n=6), it was excluded. Linear interpolation was then used to fill in any missing data within the remaining segments, as shown in Figure 1c.

Due to the controlled experimental conditions, there were few noise spikes in the Auckland FHR signals and we considered noise removal to be unnecessary. Retaining natural variation in the FHR was considered important and having some noise in the inputs also helps to regularise the neural network models that are to be used [29]. For each deceleration, the minimum FBP was determined as the 3rd percentile of the last 135 s of each segment, to ensure that any low FBP values resulted from this occlusion and not the previous one. For decelerations where the 3rd percentile was <10 mmHg (n=20), this was considered abnormally low or was observed to be caused by signal noise. In this case, the minimal FBP value was recalculated as the 20th percentile FBP during the deceleration. An example of the FHR and FBP traces marked with start locations for the FHR segments and minimum FBPs experienced during the occlusions are presented in Figure 1a,b.

Decelerations were discarded where the change in FBP between consecutive occlusions was more than 18 mmHg (n=13), a value that was tuned to ignore spurious values due to noise. They were also discarded if the minimum FBP was found to be <10 mmHg (even after re-calculation) or >80 mmHg (n=2), as these values are outside the expected range. The traces of all sheep were visually checked to ensure the most accurate minimum FBP was detected without excluding too many of the decelerations.

### 2.3. Data Pre-Processing

A total of 2698 occlusions from the 57 sheep are included in the dataset, as detailed in Table 1. Neural networks are often trained on datasets of hundreds of thousands to millions of records. We experimented with different approaches to increase the size of the dataset artificially, a strategy called ‘data augmentation’. Data augmentation through translation or by duplicating records with added uniform noise of different scales was not found to improve the accuracy of predictions during initial development of the model. We also noticed that models trained on accurately located decelerations were unable to predict FBP for time-offset FHR segments, where the deceleration started before or after the start of the segment. It is important for the model to handle cases for which the start time is less certain, as this is likely to occur when automated methods are used to locate the decelerations. Therefore, we constructed an augmented dataset consisting of 17 segments for each occlusion (as demonstrated in Figure 1b). These start up to 16 s before the true starting time and then at 2 s intervals until 16 s after the located start time. The result is a dataset containing 17 times as many segments, with 45,866 in total.

The detection of dangerously low FBP is of particular interest to this study and there were fewer occlusions where this occurred than there were where FBP was at normal levels. The segments that corresponded to a minimum FBP <35 mmHg were, therefore, duplicated in the training and validation sets. Records of each of these segments were duplicated up to a maximum of five times, or until the next duplication would have resulted in more records below the threshold than above.

Neural networks operated more effectively when the input values were small, so standardization was applied. For each segment, we subtract the corresponding baseline FHR and then divide by 47.7 bpm, which represents the overall standard deviation of FHR for all segments of all sheep. An example of a normalised segment can be seen in Figure 1d.

### 2.4. Model Testing

The FHR segments were processed using a convolutional neural network (CNN) consisting of convolutional layers with max pooling and batch normalisation after each [30]. The first convolutional layer detects shapes in the FHR segments and subsequent layers detect features consisting of combinations of these shapes. The output from the final convolutional layer is then fed into one or more fully connected neural layers. These assign weight to the different features that are found and these weighted features are used to predict the minimum FBP corresponding to each FHR segment. Various architectures were considered and tested by varying the number and type of layers and the associated hyperparameters, such as the number of convolutional filters and their sizes or the number of neurons in the fully connected layers (see Appendix A for further details).

To determine the accuracy of the model during optimisation, it was important not to use predictions for data that the model had used for training. Therefore, a test set was extracted, kept separate from the other data (see Figure 2) and used to calculate accuracy metrics by comparing the predicted FBP to the measured FBP after each model was trained: the mean absolute error, δ, mean absolute percentage error, $\delta_{\%}$ and coefficient of determination, $R^{2}$. During this process of model optimisation, the test sets consisted of all FHR segments from eight sheep, with two sheep selected randomly from each experimental group (Table 1).

After the selection of a test set, a validation set was extracted from the remaining data (Figure 2). During training, at specific intervals, the validation set was used to assess the accuracy of the model for data that had not directly influenced the training process. The training was halted when the accuracy of predictions for the validation set did not improve for 50 training epochs. This standard training procedure prevents the model from losing the ability to generalise (‘over-fitting’), i.e., learning too well the training FHR segments and losing the ability to predict for unseen FHR segments.

The non-test data were binned by sheep and by minimum FBP before being split into training and validation sets with a 5:1 ratio of records, respectively. This binning ensured that both sets contained representative samples of the data, with both containing similar quantities from the sheep and similar FBP distributions. The located and time-offset FHR segments for the same deceleration were kept in the same set during the split of training and validation to prevent information learned from one segment from reducing the error on predictions for a very similar segment in the validation set.

The accuracy of the model can vary depending on the records selected for the validation and testing sets. Every model architecture and hyperparameter selection was, therefore, trained multiple times, each time using a different split of training, validation and test sets. The test set predictions from each training run were then combined and used to calculate overall metrics to assess the predictive ability of each model configuration (Figure 2).

Examples of the mean absolute percentage errors for the training and validation sets during the training process are presented in Figure 3. The red line represents the point at which the error on the validation set is at a minimum (9.8% in this example) and is the point at which the model is saved. After this, the model begins to ‘over-fit’ the data, with the validation error staying the same or even increasing, while the training set error is still reducing.

### 2.5. Final Configuration

The optimal model configuration (Figure 4) that produced the best predictions was used to produce the results displayed in Section 3. This configuration was trained using the ADAM optimiser [31] with a learning rate of $1\times 10^{-4}$ and using the mean absolute percentage error as the loss function. This was chosen over the commonly used mean squared error to make the loss penalty larger for errors on the predictions of the lowest blood pressures (as these are of the highest interest). Rectified linear unit (ReLU) activation functions [32] were used on all layers, including the output layer, which was also initialised with a bias of 40 in an attempt to bring initial predictions close to the mean FBP value.

In the discussion and figures of Section 3, the predicted FBPs for each sheep are calculated with a model trained and validated using the FHR segments from all of the other sheep. This ensures that the predictions are made by a model that was not trained using the FHR segments from the sheep that we are assessing. The validation split was similar to that described in Section 2.4, but the selection from the binned records was done sequentially rather than randomly. This was done to maintain maximum consistency between the training and validation sets for each model. However, it is not feasible for them to be identical since a different sheep was selected for the test set each time. The overall metrics for all sheep were then generated by combining the results from the model for each individual sheep.

### 2.6. Oxford Dataset

The Oxford dataset consists of 58,488 CTG recordings made at John Radcliffe Hospital, Oxford, UK. These were recorded between April 1993 and December 2011 from singleton pregnancies of gestation at ≥36 weeks. All of these pregnancies were high-risk, where intrapartum CTG monitoring was used as per standard clinical care. Births that involved metabolic disorders, breech presentation and congenital problems were excluded from the dataset.

The final 90 min of FHR signals from this dataset were analysed and 608,177 decelerations were located using automated methods. No decelerations were extracted from 5043 of these FHR signals, either because they did not contain any decelerations or due to missing data or noise. For each located deceleration, a 150 s FHR segment was extracted. Segments were discarded if more than 50 s of data were missing and any missing data in the remaining segments were linearly interpolated. Standardisation was applied by subtracting the concurrent baseline FHR calculated over a 15 min window around the deceleration. To maintain equivalent gradients in the deceleration and the recovery of heart rate, the segments were then divided by the same standard deviation value that was used to standardise the animal data.

A model in the final configuration (see Figure 4) was trained on segments from all 57 sheep without an extracted test set and used to predict FBP from the human FHR segments. The records were split into 5279 segments from 451 cases for which severe fetal compromise occurred and 602,898 segments from 52,994 births where the outcome was not severe. Severe fetal compromise is defined as a composite outcome that includes one or more of the following: stillbirth, neonatal death, neonatal seizures, neonatal encephalopathy, hypoxic-ischaemic encephalopathy, intubation or cardiac massage, followed by 48 h in a neonatal intensive care unit. The distributions of predicted FBP from each group were then compared to explore the applicability of the model to human input data.

## 3. Results and Discussion

### 3.1. Overall Predictions for the Auckland Dataset

The predictions of minimum FBP for all FHR segments in the augmented Auckland dataset are shown in Figure 5. For each occlusion, there are 17 predicted values that appear as a vertical column of points, one for each time-offset segment. The mean absolute error between the predictions for the located and time-offset segments over all occlusions for all sheep is 1.9 mmHg with a standard deviation of 2.0 mmHg and a maximum error of 18.4 mmHg. Despite some outliers, the predictions for time-offset FHR segments are, therefore, generally in close agreement with the predictions made from the located segment.

The blue lines in Figure 5 mark the median and interquartile ranges of the predictions for 5 mmHg wide windows of the measured minimum FBP during the decelerations. These tend to lie close to the red zero-error line between the measured FBP of 30 to 50 mmHg but the predictions tend to be less accurate outside this range. The results are skewed, such that lower measured values tend to be over-predicted while the higher measured values tend to be under-predicted.

The models were able to predict the minimum FBPs in all 57 fetal sheep with a mean absolute error of δ=6.7 mmHg. The mean absolute percentage error was $\delta_{\%}=17.3$%, equating to errors of 3.5 mmHg at the lower end of the measured minimum FBP and 12 mmHg at the higher end. The coefficient of determination $R^{2}=0.36$ indicates that approximately 36% of the variation in FBP is explained by the modelled interpretation of the FHR segments. If low blood pressure is defined as being below a specified threshold, then we can assess the model based on the sensitivity, as shown in Table 2. If the model could be shown to provide accurate predictions for human fetuses, then thresholding the results in this way could provide a warning for medical practitioners during labour.

### 3.2. Predictions for Individual Sheep

The distribution of minimum fetal blood pressure during the decelerations for each sheep is presented in Figure 6 in the form of a box plot. The grey boxes show the distribution of the measured minimum FBP and the blue boxes show the distribution of minimum FBP predicted by the models. As mentioned in Section 3.1, the model tends to over-predict for those sheep with the lowest FBPs and under-predict for those with the highest FBPs.

The average accuracy of the predictions for a given fetal sheep depends on the range of FBP during the experiment. Sheep from the different experimental groups tended to have different ranges of blood pressure based on the frequency and duration of the occlusions and whether they were chronically hypoxic or not. It is likely that training a model only on similar sheep would increase the accuracy of predicting minimum FBP for those sheep. However, this would also bias the model towards similar sheep and lead to a less general interpretation of FHR segments.

Figure 7 shows examples of the FHR and FBP of four sheep presented in the order of the accuracy of the predictions made. A mean absolute error $\delta_{\%}$ of less than 10% was only found for 9 of the 57 sheep, as for the example in Figure 7a. The example shown in Figure 7b is from one of the 26 of the sheep for which $\delta_{\%}$ is between 10 and 20%. The model predicts FBP to be around 35 to 40 mmHg throughout the experiment, despite slightly higher FBP for the first few decelerations. Figure 7c shows a trace for one of 14 sheep, where $\delta_{\%}$ is between 20 and 30%. The first few decelerations are again predicted inaccurately by the model but the predictions then tend to follow the decline in minimum FBP well. However, about halfway through the experiment the FBP continues to drop, while the model continues to predict higher than measured FBP. There were only 8 sheep for which $\delta_{\%}$ was larger than 30% and an example from this group is shown in Figure 7d. The decelerations for this animal tended to end with an overshoot of FHR before dropping to a baseline level and the minimum FBP tends to be predicted with poor accuracy by the model for sheep where these overshoots occur.

### 3.3. Interpretation

The importance of the accountability [33] and interpretability [34] of neural network models has been highlighted in recent years, especially for their use in the medical field. In the present study, the inputs are short one-dimensional time series, so a fairly simple analysis can provide some insight into the workings of the model.

Figure 8 shows the averaged form of decelerations for four ranges of predicted and measured FBP. The lowest minimum FBPs tend to be predicted by the model for the FHR segments with the deepest deceleration of heart rate and those for which the deceleration occurs over the longest period of time. The presence of this association in both the measured and predicted minimum FBP supports the findings of Bennet et al. [6] and suggests a link between the deceleration shape and minimum FBP during an occlusion, where low minimum FBP suggests that fetal compromise is likely to occur.

For the hypoxic sheep (group H1-5) and the normoxic sheep that experienced 2 min occlusions (group N2-5), the decelerations also increased in depth with decreasing measured minimum FBP (see Figure 8d,f, respectively). The difference between the deceleration depth of the lowest and highest FBP groups appears to be more exaggerated for the predictions made by the model (see Figure 8c,e). This is likely due to the model learning a relationship between the minima of the FHR segment and the FBP, from the larger numbers of sheep in the other two groups. Another feature common to groups H1-5 and N2-5 is an overshoot of heart rate after the end of the occlusion. The overshoot is most pronounced when the FBP is high and the model appears to have recognised this relationship for the sheep of group N2-5 where this feature is most pronounced. For group H1-5, the model does not appear to have made this association. Westgate et al. [35] identified that large overshoots only tended to occur after total occlusions of the umbilical cord for 2 min or after prolonged periods of repeated total occlusions for 1 min.

Taking the FHR segments for the same ranges of measured FBP and then grouping them by the absolute error on the predicted blood pressure provides further information about the predictions of the model (see Figure 9). The predictions with the largest absolute errors (of 15 to 35 mmHg) tend to be for the shallowest decelerations in the lower FBP groups and for the deepest decelerations in the higher FBP groups. The averaged decelerations for the 30 to 40 mmHg and >50 mmHg groups also clearly contain a large proportion of decelerations for which overshoots occur.

### 3.4. Analysis of the Oxford Dataset

The analysis of the predictions made by the model for the Oxford dataset is more challenging due to the lack of fetal blood pressure measurements in the human data. The dataset was split into births with severe compromise and those without, as defined in Section 2.5. The model predictions for the severely compromised group are, on average, 0.5 mmHg lower than for the group where severe compromise did not occur (see Figure 10). The upper-quartile and 95th percentile values of predicted FBP are similar to those of the non-severely compromised group, but the lower-quartile and 5th-percentile values are 0.85 mmHg and 1.5 mmHg lower, respectively. While these differences between the two groups are small, they are found to be statistically significant with $p=2.1\times 10^{8}$ for an independent Student’s T-test and $p=2.4\times 10^{6}$ for a Mann–Whitney U-test, as carried out using the Python SciPy module.

The lower average of predicted FBP for the severely compromised cases indicates that the model is finding more low blood pressures for decelerations in the severe group, as would be expected. The larger range of the outliers in the non-severely compromised cases is unsurprising, given that there are 47,715 more cases in this group. The lowest outliers of predicted FBP, especially those below 20 mmHg, are likely anomalous predictions but could also be accurate and relate to single contractions that did not result in long-term harm or fetuses that experienced harm but were not classified as severe.

There are many factors affecting the relationship between FBP during single decelerations and the severity of the birth outcome, such as infection, placental defects or intrauterine growth restriction. In Figure 11, the predicted FBP is compared to other metrics that allow for a less ambiguous assessment of the predictive ability of the model. The predicted minimum FBP during the deceleration reduces as the number of previous decelerations during the final 90-min period of the CTG trace increases (see Figure 11a). The lower FBP is what would be expected after a fetus has experienced many consecutive decelerations and the average prediction drops by 5.5 mmHg, between 0 and 50 prior decelerations.

It is also expected that lower FBP would be seen towards the end of the CTG traces when a fetus has been undergoing the stresses of labour for the longest period of time and contractions are likely to be at their highest frequency. The model appears to detect such a signal and predicts lower minimum FBP during decelerations that are closest to the end of the CTG trace (see Figure 11b). The average prediction for decelerations at the end of the trace is 2.9 mmHg lower than those 90 min from the end. This provides some more confidence in the ability of the model to infer characteristics of the decelerations in human FHR that indicate the corresponding minima in FBP.

## 4. Conclusions

The model was often able to predict the minimum fetal blood pressure experienced by near-term lambs during complete umbilical occlusions of 60 to 120 s from 150 s of fetal heart rate segments with high accuracy. There was a tendency to predict close to the mean value of the minimum FBP experienced during the occlusions. This suggests that the model was not always able to extract enough information from the FHR to predict the highest or lowest FBP to a high degree of accuracy.

The accuracy of the predictions was equivalent to a similar model that was trained as part of this work (see Appendix A for further details) to predict the minimum FBP from FHR segments that were 10 min in length. This suggests that the relationship between FHR and FBP can be determined using a minimal amount of data and analysing a single deceleration can provide as much information about this relationship as the analysis of several consecutive decelerations. However, the accuracies of the predictions for both 150 s and 10 min models were improved by including additional human-inferred characteristics of the FHR (see Appendix A). This shows that while neural networks are powerful tools, it is important not to abandon metrics that research has identified as being of importance. Our results suggest that the combination of human inference and deep learning can provide better results than using either in isolation.

The accuracy of the predicted minimum FBP during decelerations varies between the different fetal sheep analysed in this study. This is partly due to the tendency of the model to predict closer to the mean FBP. Sheep with the highest or lowest FBP will tend to have less accurate predictions. However, the error on the predictions also tends to increase for animals that experience overshoots in the recovery of FHR after the end of the occlusions. This may suggest that the underlying reason for these overshoots leads to a reduction in the relationship between FHR and FBP.

When applied to human data from the Oxford dataset, the results are more difficult to interpret. Without human FBP data, it is not possible to confirm the accuracy of the predicted blood pressure. The difference in predicted minimum FBP during decelerations between the births that were defined as severe and those that were not is small, but still statistically significant. The 5th percentile of the predictions for severe cases is also 1.5 mmHg lower than for the other cases. The non-linear relationship between FBP and the likelihood of fetal compromise means that even these small changes at the lower end of the scale can lead to serious consequences. The model also tended to predict lower FBP for decelerations near the end of labour and those that occurred after larger numbers of prior decelerations. These small differences between the two groups provide encouraging evidence that the neural network used a relationship learned from the sheep FHR segments in the Auckland dataset that transferred to human FHR segments in the Oxford dataset.

One limitation of using animal experiments as training data for the model is the nature of the occlusions. The umbilical cord of each sheep was occluded completely for 1 or 2 min at regular time intervals. Decelerations in human labours can be shorter and are much more variable in depth and duration, with less regular intervals. Many of the decelerations captured from the Oxford dataset were also spaced closely enough that multiple decelerations were present within the 150 s segments used in this study, something unseen by the model during training on the animal data.

The training of the model created here would, therefore, have benefited from more variations in the length and spacing of the occlusions in the training data. Partial occlusions of different magnitudes could also provide more variation in the depth and shape of FHR decelerations, to enable the model to learn more general features that relate FHR to FBP. With more experimental data and further development and testing, such a model could be used to predict FBP and the potential for fetal compromise from FHR signals collected by CTG during human labours.

## Figures and Tables

**Figure 1 bioengineering-10-00775-f001:**
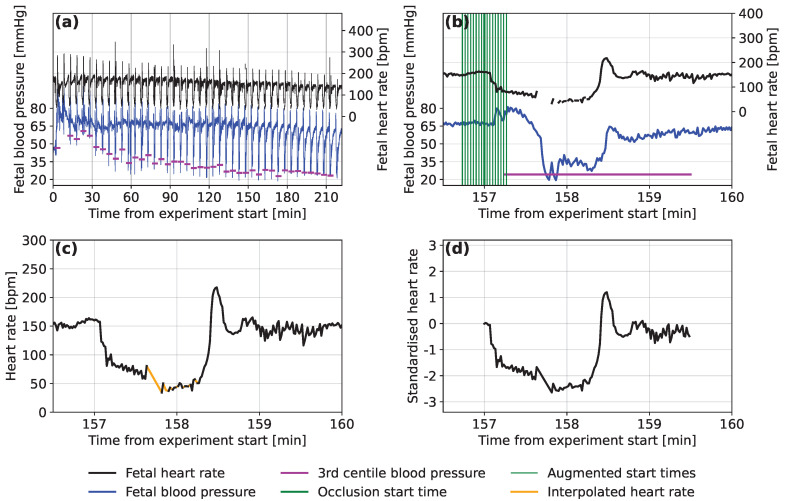
Examples of (**a**) the fetal heart rate and fetal blood pressure traces over the course of an experiment on a single subject, (**b**) a single occlusion from the same trace with start positions and the calculated minimum blood pressure marked, (**c**) a segment of the raw fetal heart rate trace for the same occlusion with the missing data filled in using linear interpolation and (**d**) the normalised form of the same segment of the fetal heart rate trace.

**Figure 2 bioengineering-10-00775-f002:**
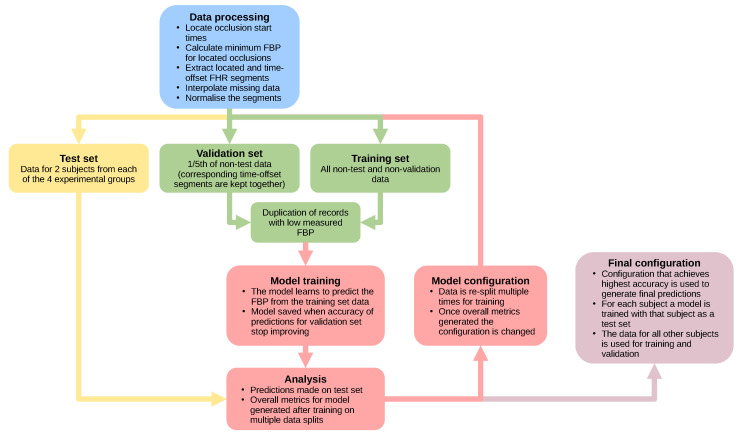
A flow chart showing the process used to handle the data and test different model architectures during the process of optimisation (FHR: fetal heart rate, FBP: fetal blood pressure).

**Figure 3 bioengineering-10-00775-f003:**
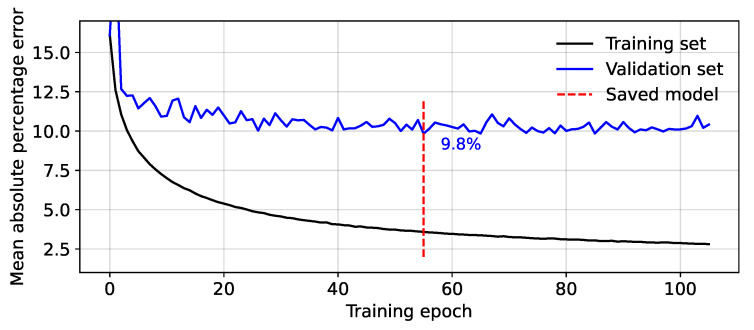
Mean absolute percentage error between the predicted and measured minimum fetal blood pressure during model training.

**Figure 4 bioengineering-10-00775-f004:**
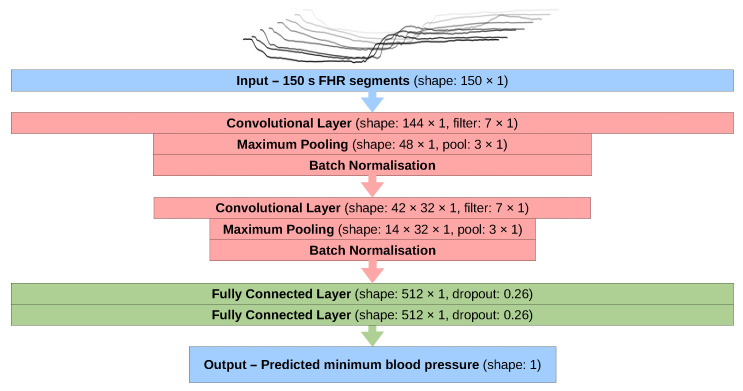
A flow chart showing the structure of the neural network model used in this study (FHR: fetal heart rate).

**Figure 5 bioengineering-10-00775-f005:**
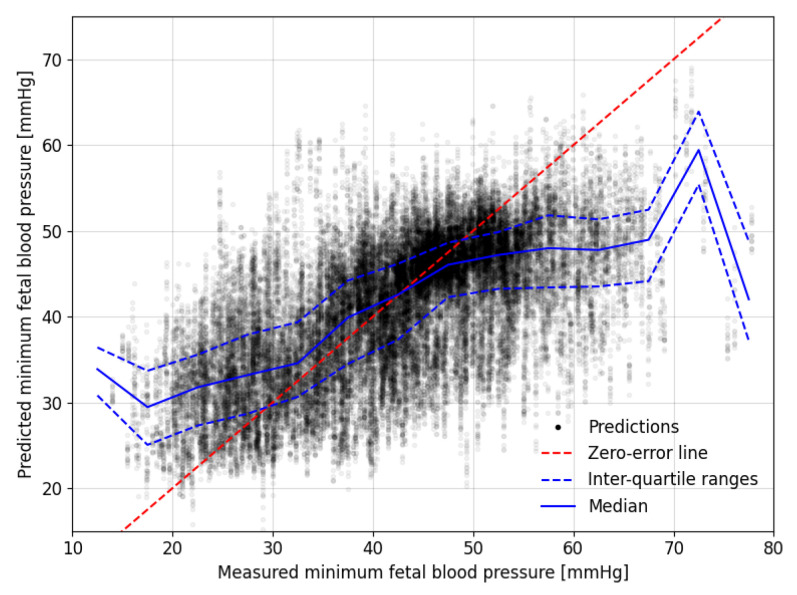
Comparison between the measured and predicted minimum fetal blood pressure during experimental occlusions. The dashed, red zero-error line shows where perfect predictions would lie. Median and interquartile range values for predictions in bins with a width of 5 mmHg of their corresponding measured minimum fetal blood pressures are shown with solid and dashed blue lines, respectively.

**Figure 6 bioengineering-10-00775-f006:**
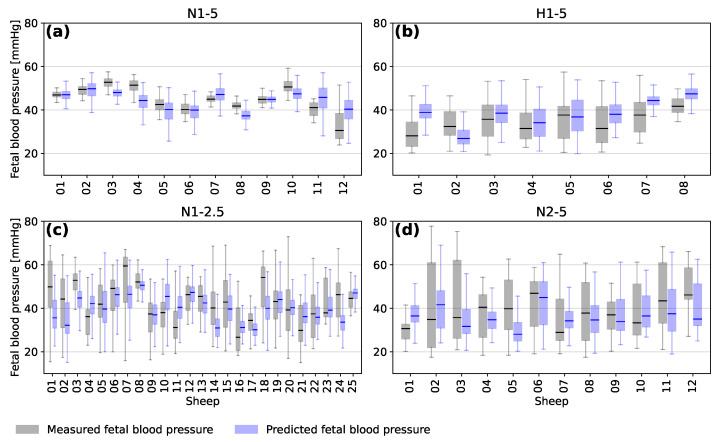
Box plots showing the median, interquartile range, minimum and maximum measured (grey) and predicted (blue) blood pressure for the occlusions of each individual subject of the four experimental groups (**a**) N1-5, (**b**) H1-5, (**c**) N1-2.5 and (**d**) N2-5.

**Figure 7 bioengineering-10-00775-f007:**
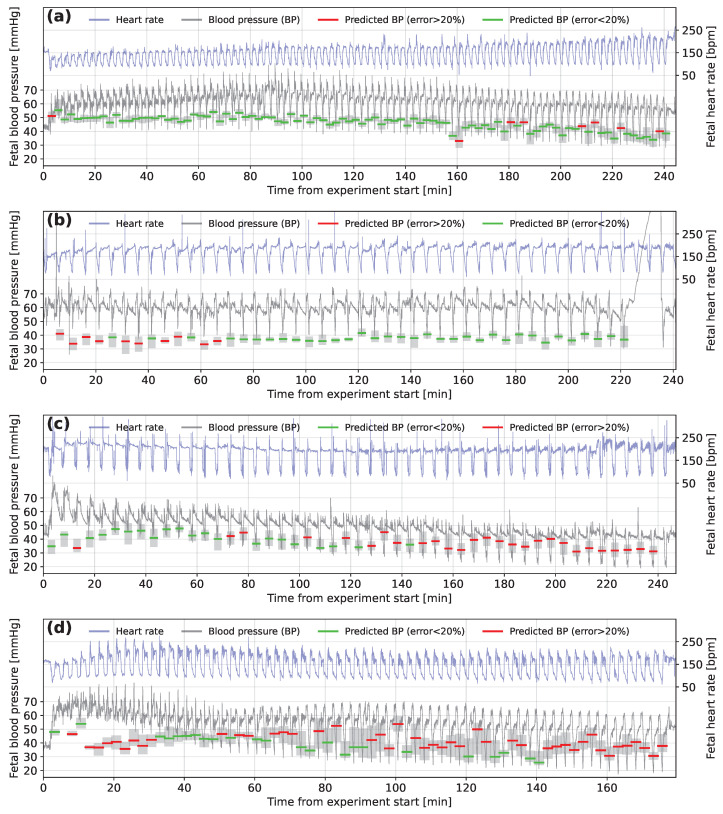
Fetal heart rate and fetal blood pressure traces marked with model predictions for subjects (**a**) N1-2.5-12, (**b**) N1-5-08, (**c**) H1-5-06 and (**d**) N1-2.5-11. The green (error less than 20%) and red (error greater than 20%) horizontal lines represent the prediction of fetal blood pressure for fetal heart rate segments located at the occlusion start time. The surrounding grey-shaded boxes show the range of predictions made on the time-offset segments for the same occlusion.

**Figure 8 bioengineering-10-00775-f008:**
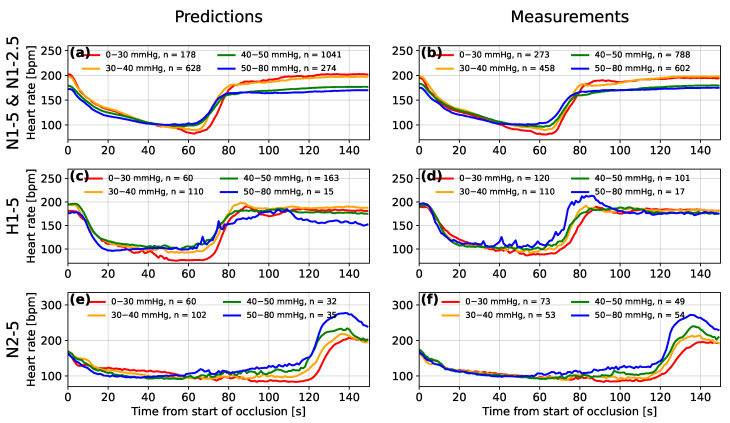
A comparison between the mean averaged fetal heart rate segments after being grouped by the range of predicted (**a**,**c**,**e**) and measured (**b**,**d**,**f**) minimum fetal blood pressure during occlusions for experimental groups N1-5 and N1-2.5 (**a**,**b**), hypoxic subjects of group H1-5 (**c**,**d**) and subjects of group H2-5 that experienced 2-min occlusions (**e**,**f**).

**Figure 9 bioengineering-10-00775-f009:**
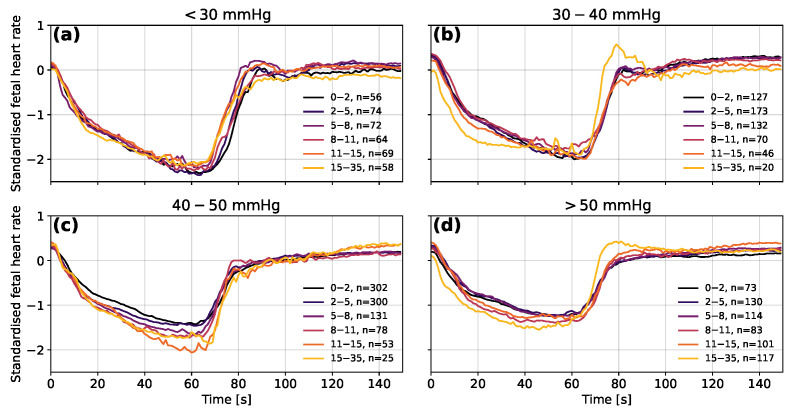
The mean averaged fetal heart rate segments grouped by the error on the predicted fetal blood pressure for measured blood pressure in the ranges (**a**) <30 mmHg, (**b**) 30–40 mmHg, (**c**) 40–50 mmHg and (**d**) >50 mmHg. Segments are shown in their normalised forms to distinguish differences in the data used as input for the model. The decelerations of group N2-5 were excluded to avoid distortions in the shape of decelerations due to the 1-min longer occlusion time. The ranges shown in the legend are the absolute errors between the predicted and measured fetal blood pressure in mmHg.

**Figure 10 bioengineering-10-00775-f010:**
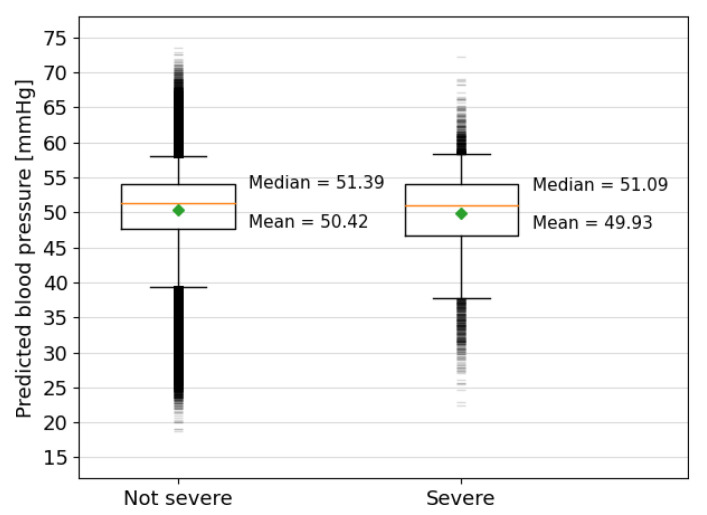
Box plots showing the predicted blood pressures for the fetal heart rate segments extracted from the Oxford dataset grouped by severity of the outcome. The orange lines mark the median values and the green diamonds mark the mean values, which are also written explicitly alongside the result of each group. The top and bottom of the boxes show the interquartile ranges, the capped vertical lines show the limit of results within the 5th and 95th percentiles and the small horizontal lines above and below mark the outliers beyond this range.

**Figure 11 bioengineering-10-00775-f011:**
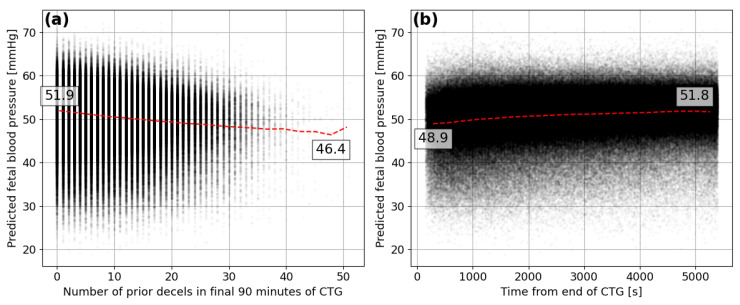
Predicted minimum FBP for each deceleration compared to the (**a**) number of prior decelerations within the final 90 min of the CTG and the (**b**) time between the deceleration start and end of the CTG in seconds.

**Table 1 bioengineering-10-00775-t001:** Descriptions of the four experimental groups.

Reference	Number of Sheep	Condition	Occlusion Length	Occlusion Spacing	Number of Occlusions	Segments in Final Dataset
			[minutes]	[minutes]		
N1-5	12	Normoxic	1	5	552	9384
H1-5	8	Hypoxic	1	5	348	5916
N1-2.5	25	Normoxic	1	2.5	1569	26,673
N2-5	12	Normoxic	2	5	229	3843

**Table 2 bioengineering-10-00775-t002:** Sensitivity and specificity of low fetal blood pressure detection given different fetal blood pressure thresholds.

Threshold	True	True	False	False	Sensitivity	Specificity
[mmHg]	Positives	Negatives	Positives	Negatives	[%]	[%]
30	2845	35,790	2154	5077	35.9	94.3
35	7419	29,495	4267	4685	61.3	87.4

## Data Availability

The data presented in this study are available upon request from the corresponding author. The Oxford data in particular are not publicly available due to data protection limitations.

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
