# Peer review of "Prediction of Fetal Blood Pressure during Labour with Deep Learning Techniques"

_bioengineering, 2023, doi:10.3390/bioengineering10070775_

Round 1

Reviewer 1 Report

This paper shows an interesting AI model in detecting fetal blood pressure (FBP). This is an important clinical issue. However, some improvement are necessary to achieve publishable standards.

1. Computational methods has been developed to estimate FBP in some pilot studies (e.g., https://www.jacc.org/doi/10.1016/j.jacc.2015.06.991). Especially, compared with low-dimensional models (e.g., Windkessel model), the computational fluid dynamics (CFD) simulation enables the detection of various hemodynamic parameters at any site in the vessel. Please comprehensively review the state-of-the-art research and enrich the introduction of the first paragraph.

2. Lines 38-39. “Fetal blood pressure monitoring could provide a more direct measure of blood flow reaching the brain.” Further clarification is needed. Actually, the cerebral blood flow is regulated by many mechanisms/factors, where anatomic variations in the Circles of Willis is a major consideration (Refer: 10.1109/ACCESS.2020.3007737). Meanwhile, the blood pressure (BP) could directly change the flow rate of cerebral arteries. Therefore the patient-specific BP value plays a key role in accurately simulating cerebral blood flow (Refer: 10.1007/s00701-022-05455-9).

3. Lines 56-59: This paragraphs is about methods and materials. Should be moved to the second section.

4. It was mentioned in additional biomarkers are needed. So the ability of deep learning to automatically extract deep features in different domains should be highlighted.

5. In Introduction please clarify the research gap that is directly related to the proposed method.

6. The fonts in the figures, especially Figures 1 and 2, are too small to read. Make sure they can be clearly shown on a A4 size screen.

7. Lines 207-208. Need clarification. Why random division was not used for testing and validation sets? “This means that the training and validation set were as similar as possible for the training of the model used for each sheep.” Is there any justification for this?

8. In figure 6, I would like to suggest marking the significant differences.

Overall it is well written but some sentences need to be restructured for better readability.

Reviewer 2 Report

i am grateful for the opportunity to have reviewed your manuscript.

i appreciate that you are upfront and candid about the limitations of the study, and understand that you are laying the foundations for further work.

despite the complexity of the study design and its methods, sufficient care has been taken by you to structure the manuscript and to elaborate when necessary. the clarity of the figures does help in this latter regard.

on the subject of the figures, choosing to colour-code Figure 2 was a prudent design choice. the colours are self-evident but the figure as a whole might benefit from a key.

of (slightly) more concern is that Figures 2 and 4 use similar colours. but the colours do not necessarily refer to similar concepts. this would not normally be a problem, but for the fact that Figures 2 and 4 are so proximal in the layout of the manuscript. 

i shall leave it to your good selves to decide if my preceding two points are valid. 

thank you, again.

Reviewer 3 Report

General comments

The submitted paper describes an attempt to predict FBP (fetal blood pressure) from FHR (fetal heart rate) data obtained by CTG during labor.  Using directly measured values of FHR and FBP in near-term pregnant sheep, the authors attempted to create a computer program, for which deep learning technologies were employed, for predicting FBP based on FHR measured by CTG.  The authors concluded that prediction was not perfect for the sheep data while applicability to human cases requires further studies using a larger and more variable training dataset, which could yield higher accuracy.   In this sense, the paper is an interim report with a limited impact. 

Specific comments

1.  A strength of this manuscript is detailed description and analyses of the work done so far.  The authors are honest and critical when describing approaches, results, analyses and interpretations.  The text is written clearly, and it is easy to follow how the study was planned and performed.

2.  In spite of the clarity of writing and the care taken to analyze data, it is disappointing to see that the deep learning using the existing data and approach did not yield a result that can correctly predict FBP for the sheet data, which is a must for the human application of the program.  I wonder if FBP can be accurately predicted by the “overall” changes in FHR.  These two mechanical characteristics are related, but the relationship may be modified by other factors, such as humoral factors (in both fetus and mother), blood chemistry, other physiological parameters and so on.  It is also possible that certain aspects of FHR changes are better correlated with FBP.  Indeed, the authors cite a work by Bennet et al. that indicates this possibility.

3.  Even if FBP can be accurately predicted, the paper does not provide how such information can be used to save serious cases of fetal cardiovascular decompensation and acidemia.  It would be informative if a bigger picture of the study is presented in Introduction or Discussion.

4.  The authors acknowledge that further work is necessary, but they do not suggest the direction of future attempts.  Please describe one or two specific and most feasible approaches that might improve predictability. 

5.  While the predicted FBP difference between the normal birth and the severely compromised birth was statistically significant, the difference in actual values was small; i.e. it was difficult to assign a threshold value for clinical problems.  In practice, I wonder if a danger threshold could be set well above the value for the severe cases such that a portion of normal range is included.  Regardless of compromised births, all such cases might be treated in the same manner clinically (provided that there are no side effects of such treatment).

6.  Although the paper is well-written, I wonder what readers can gain from this not-fully-successful approach.  What do authors think of the scientific contribution of presenting this interim report?  Wouldn’t it be better to publish a finished study?

7.  For the sheep data, certain types of noise were removed while the other was kept.  I do not believe this would change the outcome, but shouldn’t noises be treated equally?

Round 2

Reviewer 3 Report

I have no further comments.